# Coxsackievirus B3 Infection of Human iPSC Lines and Derived Primary Germ-Layer Cells Regarding Receptor Expression

**DOI:** 10.3390/ijms22031220

**Published:** 2021-01-27

**Authors:** Janik Böhnke, Sandra Pinkert, Maria Schmidt, Hans Binder, Nicole Christin Bilz, Matthias Jung, Uta Reibetanz, Antje Beling, Dan Rujescu, Claudia Claus

**Affiliations:** 1Institute of Medical Microbiology and Virology, Medical Faculty, University of Leipzig, Johannisallee 30, 04103 Leipzig, Germany; janik.boehnke@rwth-aachen.de (J.B.); christin.emmrich@medizin.uni-leipzig.de (N.C.B.); 2Institute of Biochemistry, Berlin Institute of Health (BIH) and Charité -Universitätsmedizin Berlin, Corporate Member of Freie Universität Berlin, Humboldt-Universität zu Berlin, 10117 Berlin, Germany; sandra.pinkert@charite.de (S.P.); antje.beling@charite.de (A.B.); 3DZHK (German Centre for Cardiovascular Research), Partner Side, 10115 Berlin, Germany; 4Interdisciplinary Center for Bioinformatics, University of Leipzig, 04107 Leipzig, Germany; schmidt@izbi.uni-leipzig.de (M.S.); binder@izbi.uni-leipzig.de (H.B.); 5Department of Psychiatry, Psychotherapy, and Psychosomatic Medicine, Martin Luther University Halle Wittenberg, Julius-Kuehn-Strasse 7, 06112 Halle (Saale), Germany; matthias.jung@uk-halle.de (M.J.); dan.rujescu@uk-halle.de (D.R.); 6Institute for Medical Physics and Biophysics, Medical Faculty, University of Leipzig, Härtelstrasse 16-18, 04107 Leipzig, Germany; uta.reibetanz@medizin.uni-leipzig.de

**Keywords:** ectoderm, mesoderm, endoderm, human development, embryogenesis, CAR, CXADR, DAF, self-organizing maps, genome portrayal

## Abstract

The association of members of the enterovirus family with pregnancy complications up to miscarriages is under discussion. Here, infection of two different human induced pluripotent stem cell (iPSC) lines and iPSC-derived primary germ-layer cells with coxsackievirus B3 (CVB3) was characterized as an in vitro cell culture model for very early human development. Transcriptomic analysis of iPSC lines infected with recombinant CVB3 expressing enhanced green fluorescent protein (EGFP) revealed a reduction in the expression of pluripotency genes besides an enhancement of genes involved in RNA metabolism. The initial distribution of CVB3-EGFP-positive cells within iPSC colonies correlated with the distribution of its receptor coxsackie- and adenovirus receptor (CAR). Application of anti-CAR blocking antibodies supported the requirement of CAR, but not of the co-receptor decay-accelerating factor (DAF) for infection of iPSC lines. Among iPSC-derived germ-layer cells, mesodermal cells were especially vulnerable to CVB3-EGFP infection. Our data implicate further consideration of members of the enterovirus family in the screening program of human pregnancies. Furthermore, iPSCs with their differentiation capacity into cell populations of relevant viral target organs could offer a reliable screening approach for therapeutic intervention and for assessment of organ-specific enterovirus virulence.

## 1. Introduction

Prenatal virus infections caused by members of the TORCH (Toxoplasma gondii, Others (e.g., varicella-zoster virus, parvovirus B19), Rubella virus, Cytomegalovirus, and Herpes simplex virus) group and by Zika virus interfere with human embryonal and fetal development [1]. Besides these viruses, prenatal exposure to some members of the family *Picornaviridae* and the genus enterovirus such as the coxsackieviruses are discussed to have a negative impact on human pregnancies [2,3]. Prenatal enterovirus infections could impair embryonal development through direct cytopathology or indirect mechanisms including the activation of inflammatory processes. Inflammation could interfere with neuronal development and as such contribute to psychiatric diseases including schizophrenia, autism spectrum disorder and depression [4]. The contributory role of prenatal infections to altered organ development including fetal brain injuries with a possible neuropsychiatric impact is in general difficult to assess, as these symptoms are diagnosed years after birth and therefore after the possibility to detect active virus infections [4]. Thus, the correlation of clinical symptoms that are noted after birth with maternal infection and in utero virus transmission are challenging. Moreover, Tan and colleagues raised the possibility of very early vertical transmission of viruses as shown in a mouse model for Zika virus infection [5]. Pre- and peri-implantation blastocysts are not only accessible for Zika virus by an as yet unknown mechanism, but pre-implantation virus infection could also result in fatal embryonal development [5]. This so far unknown mode of intrauterine transmission could also apply to enteroviruses and it is a matter of debate whether transplacental passage of enteroviruses could occur [6]. Clinical and epidemiological data suggests that enterovirus infections including etiologic agents of hand, foot, and mouth disease (HFMD) and members of the coxsackievirus type B group (CVB) could be an underestimated cause of adverse and even fatal pregnancy outcomes [6,7]. The case report by Ouellet et al. on the diagnosis of an intrauterine CVB3 transmission in the second trimester of pregnancy is one of the few reports on vertical transmission of enteroviruses in humans [8]. Although a life birth was noted, early neonatal death occurred [8]. This notion on rather fatal consequences of enterovirus infections during human pregnancies is supported by studies in the mouse. Vertical transmission of coxsackievirus B4-E2 (CVB4-E2) in a mouse model reduced the birth weight of the dams at a higher rate after virus inoculation at early (day 10 of gestation) as compared to later (17 days of gestation) time points of gestation [9]. Furthermore, a high rate of abortion and a lower number of offspring per litter occurred after infection at gestation day 10 [9]. This suggests that vertical transmission of coxsackieviruses appears to be associated with weight loss and an increased risk of stillbirth and abortion rather than with specific developmental disorders. This was also noted in the mouse model after CVB3 infection although a rather low fetal infection rate was present [10]. Moreover, the mouse model suggests that although the placenta is an efficient barrier against vertical transmission of enteroviruses, prenatal transmission of CVB3 occurs indicating that the placental barrier can be crossed [10]. 

In this study, we addressed CVB3 infection in induced pluripotent stem cell (iPSC)-derived cell culture models to recapitulate its possible impact on the very early steps of human embryogenesis. The relevance of such models for early prenatal virus infections at blastocyst- and early gastrulation-like stages was highlighted in a recent review in the context of the important TORCH members rubella virus, cytomegalovirus, and Zika virus [11]. CVB3 is not only a prominent cause of enterovirus-associated myocarditis and dilated cardiomyopathy [12,13,14,15], but also a discussed contributory factor to negative pregnancy outcomes [6]. Moreover, its possible association with first trimester pregnancy loss was suggested [3].

The CVB3 receptor coxsackie- and adenovirus receptor (CAR or CXADR), [16,17] is expressed in the uterus and during embryonal and fetal development as shown in mouse and rat animal models [18,19,20]. CAR-knockout mice are lethal if not otherwise rescued as shown by cardiac-restricted expression of chicken CAR in the heart, which emphasizes its essential role during cardiac development [21]. Moreover, CAR is a discussed pluripotency marker in human embryos and embryonic stem cells [22]. This suggests susceptibility of early human embryonal cells to coxsackievirus infection. This study was initiated to address CVB3 infection during very early human embryogenesis and to follow-up our previous study on the infection of three human iPSC lines with recombinant CVB3 expressing enhanced green fluorescent protein (EGFP). In these iPSC lines CVB3-EGFP infection was associated with different rates of cell death induction and influenced by the diameter of iPSC colonies [23]. Here we addressed the expression of CAR and DAF in two human iPSC lines and iPSC-derived primary germ-layer cells as a model for very early embryonal development. We asked how CVB3-EGFP infection (1) impacts the transcriptome of iPSC lines, (2) is influenced by the distribution of its receptor CAR and the co-receptor decay-accelerating factor (DAF) within iPSC colonies and (3) progresses in iPSC-derived embryonic germ-layer cells. Our data indicate that CVB3-EGFP infection of iPSC colonies is influenced by the expression pattern of the receptor CAR. Primary germ-layer cells are susceptible to CVB3-EGFP infection, which emphasizes its potential to affect early human embryonal development. So far recommendations on the management of enterovirus infections during pregnancies are lacking. Our study supports monitoring of enterovirus infections during human pregnancies to provide more clinical data on their involvement in early pregnancy loss. 

## 2. Results

### 2.1. Transcriptomic Analysis Reveals Up-Regulation of RNA Metabolic Processes in iPSC Lines after CVB3-EGFP Infection

Infection of the two human iPSC lines WISCi004-A and TMOi001-A with recombinant CVB3 encoding EGFP in its genome was followed by transcriptomic analysis to obtain a first glimpse on the course of infection. For assessment of the gene expression profile RNA was extracted from mock and CVB3-EGFP-infected iPSC lines at 24 h post-infection (hpi). At this time point of infection cell death was already induced, but at a lower level compared to later time points of infection [23]. Cell death after infection with CVB3-EGFP is induced at a faster rate on WISCi004-A iPSC line as compared to TMOi001-A iPSC line [23]. We used self-organizing map (SOM) transcriptome data portrayal (Figure 1A,B) as applied in a similar context before [24] to compare the transcriptomic pattern of WISCi004-A iPSC line with TMOi001-A iPSC line (Figure 1A) and of CVB3-EGFP-infected iPSCs with the corresponding mock control (Figure 1B). The method provides a high-resolution visualization of the transcriptome landscape of each sample. Groups of co-regulated genes are represented as colored spot-like areas where red and blue colors refer to activated and deactivated gene clusters, respectively. In the SOM portrays two spots, namely that in the left-lower and right-upper corner of the map refer to differentially expressed genes between both iPSC lines. The former spot is up-regulated in the WISCi004-A iPSC line (Figure 1A, marked with black lines). It contains genes encoding proteins involved in metal ion binding (e.g., metallothioneins (MTs), isoforms MT1A/E/F/G/H), oxidoreductase proteins, BRCA1&2 network and proteins involved in cell cycle control. The BRCA1&2 (BReast-CAncer-1 and 2) proteins are involved in cell cycle control and as tumor suppressors act in DNA repair [25]. These WISCi004-A iPSC line-specific genes involved in cellular proliferation and maintenance through DNA repair could support CVB3-EGFP replication and contribute to the faster infection and thus faster cell death rate on WISCi004-A iPSC line as observed in our previous experiments [23]. In turn, spot-genes in the right-upper corner show increased expression in TMOi001-A iPSC line. They accumulate target genes of the mammalian target of rapamycin (mTOR) network and genes related to lipid homeostasis (e.g., cholesterol biosynthesis), which points towards mechanisms of cell survival (Figure 1A, marked with green lines). The mTOR pathways contribute to cellular survival under conditions of stress including metabolic stress [26], which is posed by virus infections such as the fast replicating CVB3-EGFP. 

Next, SOM data portrayal was applied for visualization of differences in the transcriptomic pattern between mock- and CVB3-EGFP-infected iPSC lines (Figure 1B). In agreement with published data on a decrease of octamer-binding transcription factor 4 (OCT4) expression after CVB3-EGFP infection as previously detected by Western blot analysis [23], we have marked in Figure 1B a red-colored area in SOM portraits indicative for down-regulation of signature genes of pluripotency (OCT4- and NANOG-targets taken from [27]). Please note that genes are located at the same coordinates in all four portraits. As OCT4 is found near the marked area, it was deactivated in our infected samples. Another set of genes, which is typically activated after viral infections accumulates in the lower right corner of the map (Figure 1B). This area collects genes up-regulated in the CVB3-EGFP-infected iPSC lines in our study and indicates a more pronounced effect in iPSC line TMOi001-A than in WISCi004-A (Figure 1B). Among the genes that are localized to the lower right corner (Figure 1B, marked in blue) are heterogeneous nuclear ribonucleoproteins (HNRPNs), RANBP1, and RNA polymerase II subunit I and H (POLR2I and POLR2H). RANBP1 is a RAN GTPase binding protein. RAN is involved in nucleocytoplasmic transport and mitotic-spindle assembly [28]. HNRPNs are RNA-binding proteins and a trans-activating factor for the internal ribosome entry site (IRES) and as such HNRNP A1 activates human enterovirus 71 RNA translation [29]. Hence in both cell lines gene expression related to RNA metabolism/RNA metabolic processes was up-regulated which points to a general requirement of CVB3-EGFP on the infected iPSC lines to fuel its fast RNA replication cycle. As a next step, we evaluated gene sets that are related to RNA processing induced by virus infections (Figure 1C). In both iPSC lines, CVB3-EGFP infection was associated with an up-regulation of genes within the reactome_HIV_infection and HIV_lifecycle (Figure 1C). Accordingly, genes related to RNA metabolism (reactome_metabolism_of_RNA and reactome_viral_messenger_ RNA_synthesis) were up-regulated after infection of iPSC lines with the RNA virus CVB3-EGFP. In summary, CVB3-EGFP infection reduces stemness transcriptome characteristics and activates virus-induced RNA processing in both iPSC lines studied.

### 2.2. iPSC Colony Structure Determines Initial Infection Pattern of CVB3-EGFP

Transcriptomic data of CVB3-EGFP-infected iPSC lines WISCi004-A and TMOi001-A indicates that modification of RNA metabolism in particular could support viral replication in pluripotent stem cell lines. As a next step, we aimed at analysis of early virus infection as CVB3-EGFP with its fast replication cycle results in EGFP expression already at 6 hpi. A notable feature of the initial CVB3-EGFP infection in iPSC lines is the correlation between the distribution of CVB3-EGFP-infected cells and iPSC colony size [23]. Passaging of iPSC lines with collagenase results in clumps, which after plating grow to colonies with different diameters. On colonies with a diameter around or less than 200 µm CVB3-EGFP infects just the rim region of the colony, whereas on colonies with a diameter exceeding 200 µm the center of the iPSC colonies becomes accessible for virus infection. To further characterize this phenomenon, we first examined whether this pattern changes with the applied amount of virus particles, and second, we addressed how plating of iPSC lines as single cells affects this pattern (Figure 2A and Figure 2B, respectively). Irrespective of the amount of virus particles applied to smaller colonies, at 6 hpi the rim region was the only area within iPSC colonies that was positive for CVB3-EGFP infection. However, the amount of virus applied to bigger-sized colonies affected the infection pattern. At 6 hpi the number of CVB3-EGFP-positive cells in the center increased with the amount of applied virus particles (Figure 2A). In contrast to passaging of iPSC lines with collagenase, single cell plating after Accutase treatment resulted in a rather equal distribution of CVB3-EGFP-positive cells within the iPSC monolayer (Figure 2B). Passaging of iPSC lines with collagenase generates bigger-sized colonies with more densely packed cells and more intensified cell-cell junction formation than passaging with Accutase. The viral receptor CAR is a member of the junctional adhesion molecule (JAM) family and as such involved in cell-cell interactions and the formation of cell-cell adhesion sites [30]. Thus, we applied immunostaining with anti-CAR antibodies on bigger-sized iPSC colonies to determine the expression pattern of CAR. CAR expression in the outmost cell layer was lower than in the center (Figure 2C(i)). Similar to this observation, CAR expression is especially confined to cell-cell-contact areas in differentiated somatic cells as reported for the cardiac muscle cell line HL-1 and CAR-transfected Chinese hamster ovary (CHO, subclone CHO-K1) cells [31,32]. 

As a next step we applied Western blot analysis to address the expression level of CAR in iPSC lines (Figure 2C(ii)). The predicted molecular weight of CAR is 38 kDa and due to glycosylation, a 46 kDa band was detected after sodium dodecyl sulfate-polyacrylamide gel electrophoresis (SDS-PAGE). Depending on the cell type different isoforms can be present [33]. Western blot analysis of iPSC lines revealed a molecular weight band of 46 kDa besides a band of about 35 kDa. As a reference, Appendix A provides the original image file of the Western blot analysis. The expression level of CAR was higher in iPSC line TMOi001-A than in WISCi004-A, which agrees with a higher level of CAR surface expression as previously detected by flow cytometric analysis [23]. Besides this difference in CAR expression level, the ratio between both CAR molecular weight bands differed between both iPSC lines (Figure 2C(ii)). The 46 and 35 kDa molecular weight bands obtained for CAR in iPSC lines are also present in HL-1 and CHO-K1 cells as representative somatic cell lines [31,32]. In conclusion, the initial course of infection of CVB3-EGFP appeared to be dependent on the cell organization within iPSC colonies. 

### 2.3. CAR Drives Initial Pattern of CVB3-EGFP Infection in iPSC Colonies

Plating strategy and colony size influence the arrangement of iPSCs within an iPSC colony, which in turn has an impact on initial CVB3-EGFP infection (Figure 2A). As a next step, we addressed the distribution and expression of CVB3 entry factors. Besides the receptor CAR, the entry process of CVB3 involves DAF as a co-receptor, which associates with the actin cytoskeleton and through these contacts passes the virus particles to CAR [34]. Thus, we analyzed initial infection with CVB3-EGFP in relation to the distribution of the cellular components required for its entry, namely actin cytoskeleton, DAF, and CAR (Figure 3).

Actin filaments were rather homogenously distributed within iPSC colonies (Figure 3A). A notable exception was an increase in F-actin density in the rim region, the so-called epithelial-to-mesenchymal transition zone [35], and in the center of smaller-and bigger-sized colonies, respectively. DAF expression in smaller-sized colonies was restricted to the rim region (Figure 3B). In bigger-sized colonies, DAF was distributed randomly as expression spots within the iPSC colony and the rim region, which differed from the expression pattern noted for CVB3-EGFP in the rim region of bigger-sized iPSC colonies (Figure 2A). In contrast to DAF, the distribution of CAR within iPSC colonies correlated on smaller- and bigger-sized colonies with the pattern observed for CVB3-EGFP-positive cells (Figure 2A). In smaller-sized colonies, CAR expression was restricted to the rim region, whereas in bigger-sized colonies CAR was expressed in the rim region and the center (Figure 3C). Hereafter entry blocking antibodies to CAR and DAF were applied to further elucidate their relevance for CVB3-EGFP infection of iPSC line TMOi001-A in comparison to A549 as a representative somatic cell line (Figure 3D,E). A reduction on TMOi001-A was only observed after application of anti-CAR blocking antibodies, whereas anti-DAF and anti-CAR blocking antibodies reduced CVB3-EGFP extracellular titer at 24 hpi on A549 cells (Figure 3E). No additive effect was observed after co-application of anti-DAF and anti-CAR blocking antibodies on TMOi001-A (Figure 3E(i)) and A549 cells (Figure 3E(ii)). In conclusion, the expression pattern of CAR within iPSC colonies correlated with their initial susceptibility to CVB3-EGFP. Accordingly, the application of anti-CAR blocking antibodies affected its infection rate as revealed by a reduction in extracellular CVB3-EGFP titer.

### 2.4. Primary Germ-Layer Cells Are Susceptible and Vulnerable to CVB3-EGFP Infection

As a next step we addressed susceptibility of iPSC-derived germ-layer cells to CVB3-EGFP infection at three (endo- and mesodermal cells) and five (ectoderm) days after initiation of differentiation. This time schedule was chosen, such that infection could be monitored for 48 h until the end point of directed differentiation was reached as directed differentiation by the STEMdiff trilineage kit is an end-point differentiation assay. The course of infection was monitored for both iPSC lines, WISCi004-A and TMOi001-A. Both iPSC lines were included as the course of infection of CVB3-EGFP on WISCi004-A was shown to be accelerated in comparison to iPSC line TMOi001-A [23]. A similar observation was noted for the infection rate on iPSC-derived germ-layer cells. At one day post-infection, a higher number of EGFP-positive cells was detected on WISCi004-A-derived germ-layer cells (Figure 4A(i)) in comparison to the TMOi001-A-derived counterparts (Figure 4B(i)). Among the primary germ-layer cell types, mesodermal cells were especially characterized by a high level of infection. At 48 hpi an increase of infection as detected by EGFP-positive cells was noted, which on mesodermal cells was accompanied by a substantial loss of cell monolayer integrity. These observations correlate with the amount of extracellular infectious virus particles detected in the supernatant of these samples by plaque assay. At 24 hpi the amount of virus particles produced on WISCi004-A iPSC-derived mesodermal cells was significantly higher than the one obtained on ecto- and endodermal cells. Consistent with the cell loss observed for CVB3-EGFP-infected mesodermal cells, a decrease in virus titer was detected at 48 hpi in comparison to 24 hpi (Figure 4A(ii)).

At 48 hpi TMOi001-A iPSC-derived germ-layer cells resemble their WISCi004-A iPSC-derived counterparts with the highest rate of infection on mesodermal cells (Figure 4B(ii)). Thus, infection of germ-layer cells derived from iPSC line TMOi001-A is similar to those derived from WISCi004-A, but occurs with a time delay. Accordingly, on mesodermal cells derived from iPSC line TMOi001-A the number of infectious virus particles was significantly higher compared to ecto- and endodermal cells, but this was detected at 48 hpi instead of 24 hpi as noted for WISCi004-A iPSC line. In conclusion, among iPSC-derived germ-layer cells the highest rate of susceptibility together with the highest number of extracellular CVB3-EGFP virus particles was detected in mesodermal cells. This was accompanied by an iPSC line-dependent rate in virus production and loss of monolayer integrity.

## 3. Discussion

Recommendations on the management and diagnosis of virus infections during pregnancies require knowledge of the accessibility of the developing human embryo to viral pathogens. However, it is especially challenging to provide answers for the first trimester, as the time interval between initial virus infection and onset of symptoms could be too long to identify a correlation. The fetal period of human pregnancies is better accessible for methodological characterization than the embryonal period, which in part is based on the presence of a well-developed placenta at the end of the first trimester [36], which is accessible for virus diagnostics. Vertical transmission of enteroviruses from the mother to the unborn relies on detection in amniotic fluid samples by real-time PCR or in fatal cases on examination of placental or cerebral tissue samples [6,8,37]. With our study we support the notion on a possible contribution of enteroviruses to miscarriages in early weeks of pregnancy. Our iPSC-based approach for early blastocyst- and gastrulation-like stages highlights their vulnerability to CVB3 infection. The profound replication of CVB3-EGFP in derivatives of the embryonic germ layers was associated with a high level of cytopathology, especially in the mesoderm. In agreement with this observation, in vivo clinical data suggests a more severe outcome of enterovirus infection of the mother during early pregnancy as compared to later stages. Although the analysis of pregnant women with an enterovirus infection between gestational week 16 to 37 reported the birth of healthy and uninfected infants and lack of signs of transplacental passage [38], an intrauterine infection with enterovirus 71 was confirmed for a case of a 28-year-old woman with a stillbirth at gestational week 26 [39]. Moreover, for the latter clinical case viral RNA was identified in the cord blood and in the amniotic fluid besides enterovirus 71-positive signals in frozen sections of fetal tissues [39]. 

Some aspects of CVB3 infection in cells undergoing differentiation and in somatic cells were also present in iPSC lines. As human iPSC lines two commercially available and well-characterized lines were used. They were generated by two different approaches, viral vector-dependent and -independent, as this could affect their properties. iPSC lines appear to be a subtype of pluripotent stem cells and are almost, but not completely identical to embryonal pluripotent stem cells derived from the inner cell mass of human blastocysts [40]. This needs to be taken into consideration when evaluating the impact of CVB3-EGFP infection on iPSC-derived embryonal germ-layer cells. During directed differentiation, the highest titer of CVB3-EGFP extracellular particles was achieved on mesodermal cells. This peak in viral titer was reached earlier on WISCi004-A-derived mesodermal cells. Accordingly, infection of the WISCi004-A iPSC line with CVB3-EGFP was characterized by an earlier onset of viral cytopathogenicity in comparison to the TMOi001-A iPSC line [23]. The transcriptomic analysis performed in this study suggests that differential expression of genes involved in cell proliferation and survival in both iPSC lines could contribute to different levels of CVB3-EGFP cytopathogenicity. Among those are genes involved in BRCA1 and 2 regulatory networks, which for example are suppressed during infection of the monocytic THP-1 cell line with *Mycobacterium tuberculosis* to ensure its survival [41]. Accordingly, the increased cytopathogenicity of CVB3-EGFP on WISCi004-A iPSC line could at least to some degree be associated with the in comparison to TMOi001-A iPSC line higher expression level of BRCA1 and 2. Initiation of translation in CVB3 occurs through its IRES in a cap-independent manner, but requires support by cellular RNA-binding proteins such as heterogeneous nuclear ribonucleoprotein A1 (hnRNPA1), [29,42]. We have identified an increase in the reactome metabolism of RNA in both iPSC lines after infection with CVB3-EGFP, which includes proteins involved in mRNA stability and mRNA editing and members of the hnRNP family. 

The differential expression pattern of CAR on smaller- and bigger-sized iPSC colonies appears to influence the initial infection pattern of CVB3-EGFP on these colonies. Another contributing factor could be the gradient in the expression level of pluripotency markers in colonies exceeding a diameter of 500 µm [43]. Here, the expression of pluripotency markers within the center is homogenous, but increases from the center of the colony to the rim region [43]. iPSC colonies with a diameter less than 250 µm have a homogenous expression level of pluripotency markers at a level comparable to the rim region of bigger-sized colonies [43]. Such a difference in the expression of pluripotency markers could affect receptor expression and/or initial infection with CVB3-EGFP. Although such an influence may occur, on smaller-sized colonies the initial infection with CVB3-EGFP was confined only to the outer cell layer. In our previous study on the course of infection of CVB3-EGFP on human iPSC lines we have identified through immunofluorescence analysis a lower expression level of the pluripotency marker OCT4 in some of the infected iPSCs, whereas some of the infected cells expressed OCT4 at a level comparable to the mock population [23]. The analysis of CAR expression in iPSCs revealed interesting aspects for future evaluation. Its sustained expression in iPSC lines and the influence of the method of cultivation, either as clumps or single cells, on the initial infection pattern with CVB3-EGFP reflects its proposed function as a pluripotency factor and tight junction protein, respectively [22,44]. It is tempting to speculate that the truncated version of CAR identified in our Western blot analysis lacks the 11.6 kDa cytoplasmic tail which drives plasma membrane localization of CAR [45,46]. The Universal Protein Ressource (UniProt) provides detailed information on protein sequences and annotation data. UniProt [47] lists several isoforms for CAR (CXADR, #P78310) including the reference sequence P78310-2 and the sequence P78310-5, which lacks the cytoplasmic domain and has a predicted molecular weight of 25 kDa. After glycosylation, this isoform could represent the 35 kDa band detected by our Western blot analysis for CAR. Tailless CAR localizes to the plasma membrane and enables CVB3 binding [48], although the associated membrane microdomain differs from wild-type CAR [46]. 

The high CVB3-EGFP infection rate in iPSC-derived cell culture models highlights their application in studies on viral pathogenicity and in the development of antiviral compounds. The mesoderm directs cardiac development and the ectoderm development of the brain. The possible impairment of these developmental pathways during prenatal enterovirus infection is supported by clinical data obtained through sonographic examination. Prenatal enterovirus infection was associated with cerebral ventriculomegaly and cardiomyopathy with ventricular dysfunction [49] in addition to cases of neonatal enterovirus-associated meningoencephalitis [50]. Furthermore, the presence of enterovirus 71 in neurons of the midbrain together with indications for pulmonary edema indicate impairment of central nervous system (CNS) development [39]. These clinical cases on neuronal impairment by prenatal enterovirus infections are supported by studies on its receptor CAR in the developing and adult brain. The receptor CAR is expressed in the embryonic ectoderm in the developing mouse [51], in the adult mouse [52], and human brain [53]. So far treatment options for enterovirus infections are limited and include application of monoclonal antibodies as shown for causative agents of HFMD [54] and for coxsackievirus [55] in addition to the therapeutic potential of the soluble analogue of the receptor CAR [56]. Moreover, differences in pathogenicity and neurovirulence among newly emerging enteroviruses could be addressed in iPSC-derived models. A recent study on enterovirus A71 strains that were isolated from children with HFMD during an outbreak between 2015 and 2016 in Vietnam suggests that in comparison to the subgenotype B5 the new EV-A71 subgenotype C4 is more neurovirulent [57]. 

The noted cytopathology of CVB3-EGFP infection in embryonal germ-layer cells indicates that in vivo infection of blastocyst- and early gastrulation-like embryonal stages could severely impair embryonal development. This is supported by a study conducted in Korea on 51 gravidas with birth at full-term as compared to pre-term delivery and missed abortions [3]. Here, the incidence of an enterovirus infection in the missed abortion group was significantly higher than in the group with full-term or pre-term delivery [3]. Moreover, within the missed abortion group 5 cases were demonstrated to be CVB3-positive [3]. In a mouse model of CVB4-E2 infection during pregnancy, dams were infected at different time points during gravidity starting at embryonic day (E) 4 [58]. E4 is the time point of blastocyst implantation in the mouse [59]. The viral load in the pancreas from pups of dams infected at E4 were higher in comparison to the pancreatic samples from pups of dams infected in the second or third week of gravidity. Although the study by Sarmirova and colleagues focuses on pathological aspects of CVB4-E2 infection in the pancreas, it highlights that vertical transmission can occur at very early time points of embryonal development. In future studies the possible interference of coxsackievirus infection with very early human embryonal development including blastocyst- and gastrulation-like stages need to be further evaluated through suitable in vitro protocols such as the two-dimensional micropatterning differentiation approach [60]. Human pluripotent stem cells are seeded as micropatterned colonies and differentiated into organized germ layers in resemblance of embryonic patterning [60]. 

We suggest including CVB3 infection in the epidemiological assessment of virus infections during human pregnancies as a possible cause of abortion. Furthermore, a positive test for an enterovirus infection during pregnancy should be followed by close monitoring of the developing fetus. This could provide relevant clinical data required for further recommendations on enterovirus infections during human pregnancies.

## 4. Materials and Methods 

### 4.1. Cell Lines and Cultivation

For maintenance of the vector-free human episomal TMOi001-A iPSC line (alternative name Gibco episomal iPSC line, A18945), (Thermo Fisher Scientific, Waltham, MA, USA) and the lentiviral WISCi004-A iPSC line (alternative name IPS(IMR90)-1), (WiCell, Madison, WI, USA) mTeSR™1 medium (STEMCELL Technologies, Vancouver, British Columbia, Canada) supplemented with 10 µg/mL gentamycin was used. After dissociation iPSC lines were plated on Matrigel™ (BD Biosciences, Franklin Lakes, NJ, USA), which was dispensed in Dulbecco’s Modified Eagle’s Medium (DMEM): nutrient mixture F-12 (DMEM/F-12). During cultivation all iPSC cultures, including mock- and virus-infected samples, were subjected to daily medium change. For enzymatic passage at a ratio of 1:6 to 1:10 after 3 to 5 days of cultivation collagenase type IV (Thermo Fisher Scientific) dissolved at a concentration of 1 µg/mL in DMEM/F-12 was used. During passaging Y-27632 ROCK inhibitor (Santa Cruz Biotechnology, Dallas, TX, USA) was added at a concentration of 10 μM. For virus infection experiments iPSC lines were passaged by collagenase at a ratio of 1:4 or for experiments requiring single cell suspension at a density of 50,000 cells/cm^2^ after Accutase (Sigma-Aldrich (Merck), Taufkirchen, Germany) treatment. Human lung carcinoma cell line A549 was cultivated in DMEM with high glucose under supplementation with 10% fetal calf serum and antibiotics. 

### 4.2. Directed Differentiation of iPSC Lines

The STEMdiff^TM^ trilineage differentiation kit (StemCell Technologies) as an end-point differentiation assay was applied for directed differentiation of iPSC lines according to manufacturer’s instructions. After plating of iPSCs as single cell suspension, the respective STEMdiff^TM^ trilineage differentiation medium was added and changed daily. The end-point of differentiation was reached after 5 days (mesoderm and endoderm) and 7 days (ectoderm) of cultivation. 

### 4.3. Virus Infection

For preparation of CVB3-EGFP stock virus the pEGFP-CVB3 plasmid (provided by Zhao-Hua Zhong, Department of Microbiology, Harbin Medical University, Harbin, China) was transfected into HeLa cells. EGFP as a reporter protein is encoded in the viral genome after the start codon of the viral VP4 protein. Due to the presence of a protease cleavage site, EGFP and VP4 are expressed as single proteins. Although no viral EGFP fusion proteins are expressed, the CVB3-EGFP phenotype is attenuated compared to the wild-type CVB3 [61,62]. After removal of cell debris through centrifugation at 500× *g* for 10 min and passage through a 0.45 µm membrane virus-containing supernatants were subjected to semi-purification through ultracentrifugation at 25,000 rpm for 120 min at 4 °C with a 20% sucrose cushion. Thereafter pellets were resuspended in mTeSR™1 medium and plaque assays on HeLa were used for titer determination [63]. In infection studies of iPSC lines or iPSC-derived primary germ-layer cells plated in a six well culture plate 1.5 × 10^6^ plaque forming units (PFU) of CVB3-GFP were applied per well if not otherwise indicated. After an incubation period of 1 h virus inoculum was replaced with fresh mTeSR™1 medium. For antibody blocking experiments A549 cells were infected with an multiplicity of infection (MOI) of 33.

### 4.4. RNA Isolation, Microarray Gene Expression Analysis, and SOM Portrayal

After extraction by Trizol reagent (Thermo Fisher Scientific) RNA was purified by the Direct-zol RNA kit (Zymo Research, Irvine, CA, USA) according to manufacturer’s instructions. Quality analysis of RNA samples was performed on a fragment analyzer (Advanced Analytical Technologies (Agilent), Ankeny, IA, USA). RNA of samples with a RIN (RNA integrity number as a means of quality assessment) equal to 7 or greater was further processed and hybridized to Illumina HT-12 v4 Expression BeadChips (Illumina, San Diego, CA, USA) and measured on the Illumina HiScan. Expression data was processed through SOM machine learning as described in our previous publication on rubella virus-infected iPSC lines [24]. Briefly, we applied the R-program “oposSOM” [64] which generates images of the transcriptomic landscape of each sample revealing clusters of over- and under-expressed genes as red and blue spot-like areas. Their functional context was evaluated by means of gene set analysis as implemented in [65].

### 4.5. Application of Blocking Antibodies

For assessment of receptor engagement by CVB3-EGFP, antibodies against CAR (#05-644, clone RmcB, (Merck, Darmstadt, Germany)) and DAF (#sc-133220, Santa Cruz Biotechnology) were applied at a dilution of 1:50 and 1:10, respectively. Cells were incubated with isotype control antibody at a dilution of 1:50 (normal mouse IgG1, #sc-3877, Santa Cruz Biotechnology) or blocking antibodies for 120 min followed by addition of CVB3-EGFP virus. After 60 min of incubation medium change was applied together with re-application of the respective antibody until further analysis.

### 4.6. Western Blot Analysis

Total protein lysates were generated with cytoplasmic and nuclear protein fractions by NE-PER nuclear and cytoplasmic extraction kit (#78835, Thermo Fisher Scientific) according to manufacturer’s instructions. CAR is a discussed pluripotency factor [22] and only cytoplasmic extracts were subjected to further analysis as only cytoplasmic CAR was considered relevant for CVB3-EGFP entry. Total protein concentration in the cell extracts was determined by standard Bradford assay to ensure equal loading in Western blot experiments. Unspecific binding during immunoblotting was blocked with 5% (*w*/*v*) nonfat dried milk powder in 0.1% Tween-20 in PBS. Primary antibodies were used for overnight incubation of the PVDF membrane at the following dilutions: anti-CAR (H-300, #sc-15405, Santa Cruz Biotechnology) and anti-cofilin (#sc-5279, Santa Cruz Biotechnology) at 1:200. An incubation with horseradish peroxidase-conjugated secondary antibodies was followed by detection through chemiluminescence (Western Lightning Plus-ECL, Perkin Elmer) on a C-DiGit^®^-Blotscanner (LI-COR Biosciences, Lincoln, NE, USA).

### 4.7. Immunofluorescence Analysis

For immunofluorescence analysis, iPSCs were cultivated in 24 well plates and fixed with 2% (*w*/*v*) paraformaldehyde. Staining of surface expressed DAF and CAR was carried out without permeabilization, whereas F-actin staining with Alexa Fluor 555 phalloidin (Thermo Fisher Scientific) at a 1:40 dilution was applied after permeabilization with 0.3% Triton X-100 in PBS. After blocking of unspecific binding with goat or donkey serum, anti-CAR (#sc-15405 H-300, Santa Cruz Biotechnology or clone RmcB) or anti-DAF (#sc-133220 H-7, Santa Cruz Biotechnology) antibodies were applied at a dilution of 1:100 followed by addition of goat anti-rabbit or donkey anti-mouse IgG (H + L) Cy3 MinX secondary antibodies (Dianova, Hamburg, Germany). Addition of primary and secondary antibodies and phalloidin dye was followed by application of the DNA-specific stain Hoechst bisbenzimide 33258 (Thermo Fisher Scientific) at 5 µg/mL in PBS. Immunofluorescence analysis of CAR with clone H-300 was done in reference to our previous publication [23], which showed differential surface expression of CAR on TMOi001-A and WISCi004-A iPSC. This was complemented by the application of the well-characterized RmcB clone, which was also used for antibody blocking experiments. Appendix A shows a representative flowcytometric analysis on the lack of fluorescence signal after application of the donkey anti-mouse Cy3 secondary antibody. Images were taken with an Olympus XM10 fluorescence microscope and processed for minimal alterations in contrast and background in CorelDRAW2020.

### 4.8. Statistical Analysis

Statistical analysis was based on one-way analysis of variance (ANOVA) followed by Bonferroni’s multiple comparison test done with Graph Pad Prism software (GraphPad Software, Inc., La Jolla, CA, USA). Level of significance data in diagrams is shown by asterisks (* *p* < 0.05, ** *p* < 0.01, *** *p* < 0.001). Data are shown as means ± standard deviation (SD). 

## Figures and Tables

**Figure 1 ijms-22-01220-f001:**
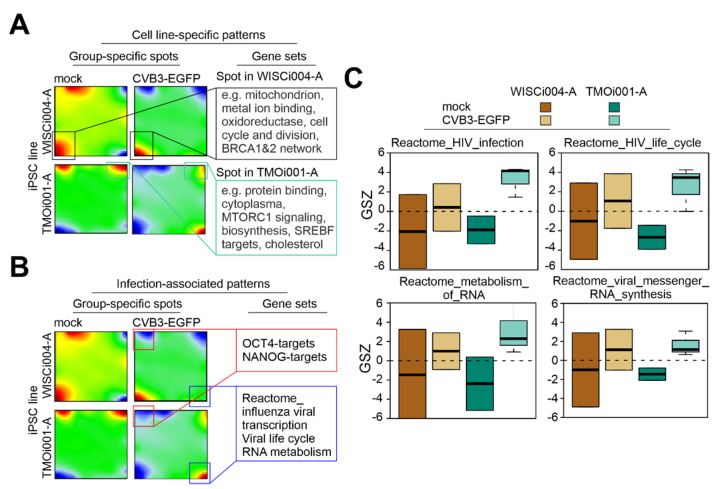
Transcriptome of CVB3-EGFP-infected iPSC lines indicates a similar down-regulation of pluripotency genes and an up-regulation of pathways involved in RNA metabolism. (**A**,**B**) SOM portraits of mock- and CVB3-EGFP-infected iPSC lines. (**A**) Spots with a differential expression between the iPSC lines (genes up-regulated in WISCi004-A iPSC line are marked with black lines; genes up-regulated in TMOi001-A iPSC line are marked with green lines). (**B**) Spots with a differential regulation in both iPSC lines after CVB3-EGFP infection are highlighted: in red lines stemness factors were down-regulated and in blue lines gene patterns associated with virus infections were up-regulated. (**C**) Gene set expression signatures that are associated with the cellular response to virus infections and with RNA metabolism and mRNA synthesis were up-regulated (in units of the gene set *Z*-score, GSZ) in CVB3-EGFP-infected cells.

**Figure 2 ijms-22-01220-f002:**
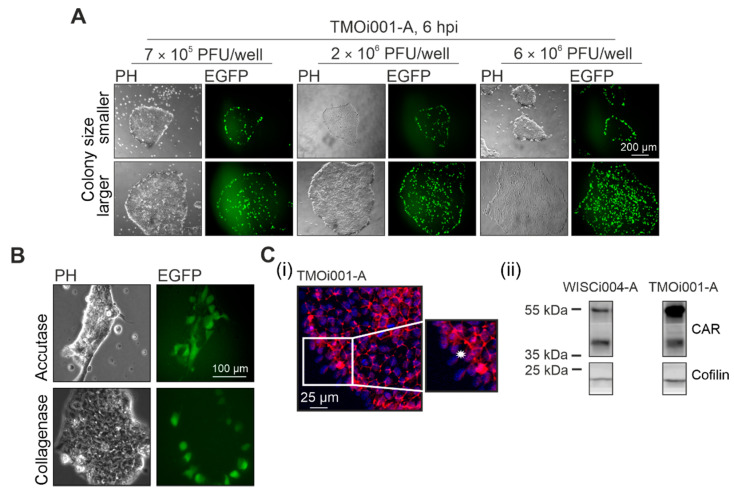
Initial CVB3-EGFP infection on iPSC colonies (**A**,**B**) and expression of its receptor CAR in mock-infected iPSC lines (**C**). (**A**) The iPSC line TMOi001-A was infected with the indicated amount of CVB3-EGFP virus particles 24 h after plating. (**B**) The iPSC line TMOi001-A was passaged with collagenase or Accutase and at 24 h of cultivation CVB3-EGFP infection was initiated. (**A**,**B**) At 6 hpi initial EGFP expression was monitored through fluorescence microscopy. (**C**) The iPSC line TMOi001-A was processed for immunostaining with anti-CAR antibody (H-300) after 4 days of cultivation (**C**(i)). 30 to 40 µg of the cytoplasmic fraction obtained after 4 days of cultivation of indicated iPSC lines was processed to Western blot analysis with anti-CAR (H-300) antibodies. Detection of cofilin was used as a loading control (**C**(ii)). Asterisks mark areas of reduced cell-cell contact and reduced CAR expression. PH, phase contrast; PFU, plaque forming units.

**Figure 3 ijms-22-01220-f003:**
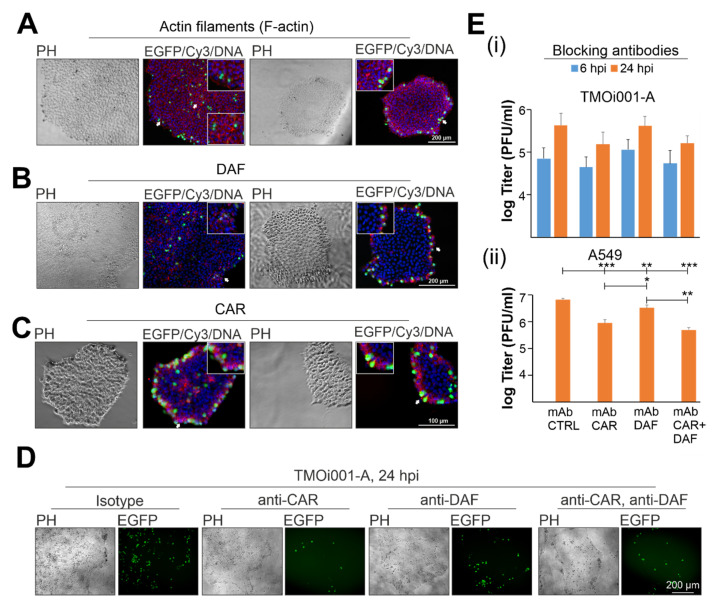
Distribution of CAR influences susceptibility of iPSC colonies to CVB3-EGFP. (**A**–**C**) At 24 hpi CVB3-EGFP-infected (shown in green) TMOi001-A iPSC line were stained for F-actin with phalloidin and monoclonal antibodies against indicated proteins (shown in red) together with a DNA counterstain (shown in blue). As anti-CAR antibody clone RmcB was used. Fluorescence microscopy analysis included smaller- and bigger-sized iPSC colonies. Arrows depict areas shown at higher magnification in the insets. (**D**,**E**) Blocking antibody experiments were carried out with monoclonal antibodies against CAR (clone RmcB) and DAF and compared to the isotype control antibody (mAb CTRL, isotype control (CTRL) monoclonal antibody (mAb)). (**D**) At 24 hpi CVB3-EGFP infection on collagenase-plated iPSC line TMOi001-A was monitored by fluorescence microscopy and (**E**) at 6 hpi and/or 24 hpi the amount of extracellular virus particles was determined by standard plaque assay for collagenase- and Accutase-plated TMOi001-A and A549 as representative somatic cells (*n* = 3 to *n* = 4, * *p* < 0.05, ** *p* < 0.01, *** *p* < 0.001).

**Figure 4 ijms-22-01220-f004:**
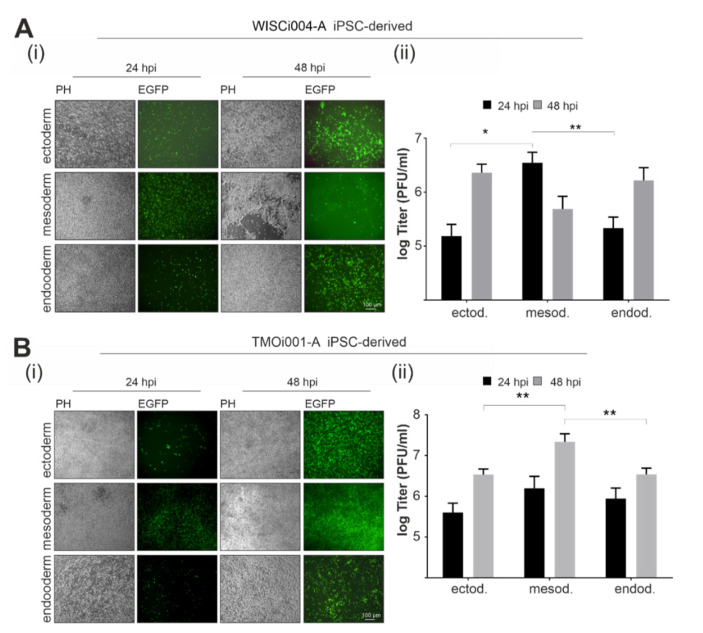
iPSC line-derived embryonal germ-layer cells are infected by CVB3-EGFP and reveal a high level of replication in mesodermal cells. The iPSC lines (**A**) WISCi004-A and (**B**) TMOi001-A were subjected to directed differentiation by the STEMdiff trilineage differentiation kit. CVB3-EGFP was applied two days before the end-point of lineage differentiation. Course of infection was monitored through (i) fluorescence microscopy to detect EGFP expression and (ii) plaque assay to monitor virus titer production at 24 and 48 hpi (*n* = 3, (* *p* < 0.05, ** *p* < 0.01)). ectod., ectoderm; mesod., mesoderm; endod., endoderm.

## Data Availability

Not applicable.

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
