# Peer review of "Coxsackievirus B3 Infection of Human iPSC Lines and Derived Primary Germ-Layer Cells Regarding Receptor Expression"

_ijms, 2021, doi:10.3390/ijms22031220_

Round 1

Reviewer 1 Report

The paper of Böhnke et al addresses an important issue of possible causes of early pregnancy loss due to viral infection. The authors use hiPS cells as a model for early embryonic development to perform viral infections that could lead to impaired development and potential loss of early pregnancy.
This paper is very timely and well executed however before accepting for publication, I would like to suggest several minor changes to be considered by the authors.

Introduction:
Since the paper is revolving around early embryonic development, more information information regarding experiments performed the mouse (or any other) model should be given. How is the embryonic development affected in infected pregnant mice? This information appears at the end of the discussion but the introduction is missing this embryology aspect.

Results
Generally hiPSCs and hESCs are both used as models for early human development despite having different origin. Pluripotent state and developmental potency these two cell types can differ (as demonstrated by many assays including teratoma assay see Nat Commun. 2018 May 15;9(1):1925. doi: 10.1038/s41467-018-04011-3). Are hESC cell lines behaving the same way as hiPSCs? I a bit worried that the authors are making too strong conclusion based only on the two particular cell lines tested. I would love to see these experiments performed on hESC lines as well. If that’s not possible that should e at least discussed.

Figure 1A There is difference in gene expression in bottom left corner in TMOi001-A cell line vs WISC004-A. What are these genes? Could this difference explain the different time needed for equal infection rates? Why different cell lines behave differently? Perhaps a third cell line should be included (normally its all done in triplets).

Overall I am missing quality controls for cells. OCT4/SSEA4/NANOG/TRA1-81 either by FACS or immunofluorescence. It is important to know the starting point for all experiments is the same given the fact that infection rates differ based one colony size. If infection is really reducing stemness a co-staining with NANOG should be done to directly show the effect. It could be that the infection is really possible only at the cells that enter EMT and that’s why the size of the colony matters. In many cases the edges of iPSC colony contain OCT4+ but already NANOG- cells and so does the middle of the colony which starts to be multilayered. Is that the case here?

Discussion
In the discussion the authors mention viral infections of pregnant mice (citation 52). This paper addresses only the issue of pancreatic development and does not analyse early embryonic development such as gastrulation or somitogenesis, which is far more relevant to early pregnancy loss. Also these authors do not mention anything regarding possible pregnancy loss. As such this citation is not in favour of the paper. It actually undermines it by showing that the pups are born.

I am also missing a follow up experiments. It would be very interesting to see whether the viral infection actually interferes with mesoderm development at the onset and during gastrulation. Of course this can only be done in mouse model, however another human gastrulation model could be used such as developed by Brivanlou and Warmflash (Nat Protoc. 2016 Nov;11(11):2223-2232. doi: 10.1038/nprot.2016.131. Epub 2016 Oct 13.) It is at least worth discussing.

Author Response

General comments and general changes made to the original manuscript

The authors want to thank the reviewers for their time spent on our manuscript. Their helpful comments and suggestions for revision improved the manuscript. In Figure 3E(ii) the level of significance for mAb DAF was corrected from ** to *.

Point-by-point response to the reviewer’s suggestions

Reviewer 1

# This paper is very timely and well executed

Thank you for this supportive comment.

#Introduction:
Since the paper is revolving around early embryonic development, more information regarding experiments performed the mouse (or any other) model should be given. How is the embryonic development affected in infected pregnant mice? This information appears at the end of the discussion but the introduction is missing this embryology aspect.

            Thank you for this valuable comment. In line 75 to 88 in the introduction we have added a case report on intrauterine coxsackievirus transmission in a human pregnancy and data obtained in mouse models. Especially the inoculation of pregnant mice at an early pregnancy stage results in an increased risk of abortion and stillbirths. These aspects support the outline of the introduction through highlighting the relevance of human iPSCs as an important platform for the study of prenatal CVB3 infection.

#Results
Generally hiPSCs and hESCs are both used as models for early human development despite having different origin. Pluripotent state and developmental potency these two cell types can differ (as demonstrated by many assays including teratoma assay see Nat Commun. 2018 May 15;9(1):1925. doi: 10.1038/s41467-018-04011-3). Are hESC cell lines behaving the same way as hiPSCs? I a bit worried that the authors are making too strong conclusion based only on the two particular cell lines tested. I would love to see these experiments performed on hESC lines as well. If that’s not possible that should e at least discussed.

The publication mentioned by the reviewer raises an important issue of human pluripotent stem cells regarding their malignant potential. This aspect needs to be considered during data discussion, but may not necessarily interfere with the data analysis. Human iPS cells and human ES cells share almost all characteristics of pluripotent cells. However, gene expression profiles of different iPSC lines are very similar but not identical to human ES cell lines. In this context questions were raised regarding the carcinogenic potential of human iPS cells. These concerns are still discussed because there are diseases where cell replacement strategies are evaluated in clinical trials in patients. Today, tools have been established to characterize the pluripotency, the developmental potential, and the genomic integrity of iPSC lines. This for example includes the PluriTest, ScoreCard and other techniques also applied in the article mentioned by reviewer 1. This article demonstrates that EB assays are not suitable to replace teratoma analysis. This conclusion was drawn with the current state-of-the-art techniques applied in the article for the characterization of pluripotency in iPSC lines. Furthermore, different techniques have been developed to improve the selection of somatic donor cells, reprogramming techniques, and culture conditions which has been reviewed by others. Nevertheless, iPSC lines are not identical to ES cells. Therefore, we have added a summary of these considerations in line 345 to 350. As a reference the publication by Chin et al., 2009, (doi: DOI: 10.1016/j.stem.2009.06.008) was added.

#Figure 1A There is difference in gene expression in bottom left corner in TMOi001-A cell line vs WISC004-A. What are these genes? Could this difference explain the different time needed for equal infection rates? Why different cell lines behave differently? Perhaps a third cell line should be included (normally its all done in triplets).

Thank you for this very helpful comment. We have addressed this aspect and analysed the iPSC line-specific spots in our SOM portraits. We have added this aspect to Figure 1A, which is described in the results section in line 136 to 156. We have identified genes, that appear to be expressed in an iPSC-line specific manner. The possible contributory role of these gene expression patterns to the differences in cytopathogenicity of CVB3-EGFP on these iPSC lines as observed in our previous study (Hubner et al., 2017) is discussed in line 354 to 361. iPSC lines were obtained from different donors and therefore they carry an intrinsic biological variability due to the variability of human beings. We used iPSC lines from different donors to include this variability in our vitro model and to reduce the possibility of a cell line-dependent effect. Both iPSC lines are commercially available and have been established and published by others representing healthy control cell lines.

#Overall I am missing quality controls for cells. OCT4/SSEA4/NANOG/TRA1-81 either by FACS or immunofluorescence. It is important to know the starting point for all experiments is the same given the fact that infection rates differ based one colony size. If infection is really reducing stemness a co-staining with NANOG should be done to directly show the effect. It could be that the infection is really possible only at the cells that enter EMT and that’s why the size of the colony matters. In many cases the edges of iPSC colony contain OCT4+ but already NANOG- cells and so does the middle of the colony which starts to be multilayered. Is that the case here?

In our previous study on CVB3-EGFP-infected iPSC lines (Hubner et al., 2017) we have addressed the association of CVB3-EGFP infection with pluripotency marker OCT4 through immunofluorescence and Western blot analysis. In comparison to the mock-infected control, the expression of OCT4 was reduced after infection with CVB3-EGFP. As revealed by immunofluorescence analysis, the infection with CVB3-EGFP was in some cells associated with a reduced OCT4 expression level. In the discussion, line 367 to 379, we have addressed the possible influence of the iPSC colony size-dependent expression level of pluripotency markers on the initial infection pattern of CVB3-EGFP. As a reference the publication by Warmflash et al., 2014, (doi: 10.1038/nmeth.3016) was added.

#Discussion
In the discussion the authors mention viral infections of pregnant
mice (citation 52). This paper addresses only the issue of pancreatic development and does not analyse early embryonic development such as gastrulation or somitogenesis, which is far more relevant to early pregnancy loss. Also these authors do not mention anything regarding possible pregnancy loss. As such this citation is not in favour of the paper. It actually undermines it by showing that the pups are born. –

Within this context raised by reviewer 1 we noted that reference [51], Sarmirova et al. and [52], Paria et al. were not clearly stated. We have revised the sentences in line 421 to 423 accordingly. The reference Paria et al. states embryonal day 4 in the mouse as the time point of blastocyst implantation. The reference Sarmirova et al highlights that vertical transmission to the embryo was possible even after application of the virus at this early time point. The pancreas is one of the preferred target organs of coxsackieviruses and experimental proof of the presence of the virus in the fetus supports vertical transmission. The time window at which vertical transmission can occur needs to be defined in the future to get further insights into early embryonal development including gastrulation and somitogenesis. We have revised the original statement to clarify this aspect. Accordingly, line 421 to 423 state that “Although the study by Sarmirova and colleagues focuses on pathological aspects of CVB4-E2 infection in the pancreas, it highlights that vertical transmission can occur at very early time points of embryonal development.”

#I am also missing a follow up experiments. It would be very interesting to see whether the viral infection actually interferes with mesoderm development at the onset and during gastrulation. Of course this can only be done in mouse model, however another human gastrulation model could be used such as developed by Brivanlou and Warmflash (Nat Protoc. 2016 Nov;11(11):2223-2232. doi: 10.1038/nprot.2016.131. Epub 2016 Oct 13.) It is at least worth discussing.       

We are very grateful for this comment, which corresponds to the previous suggestion by reviewer 1. Therefore, we added the statement regarding the suggested reference to line 423 to 428.

Reviewer 2 Report

The manuscript, ‘Coxsackievirus B3 infection of human iPSC lines and 3 derived primary germ layer cells with regard to 4 receptor expression’ by Böhnke et. al., elegantly showed why enterovirus infections during human pregnancies could be involved in early pregnancy loss. This is of broad interest. However, there are the following concerns that should be addressed:

Major comments:

  1. The placental barrier is historically known to restrict many viruses from entering the fetus.  The placental trophoblasts form the interface between the fetal and maternal environments are highly resistant to viral infections (Delorme-Axford et. al., 2013). The reference (31) that is cited by the authors shows that transplacental passage of virus does not occur readily and that most neonates of infected mothers will be unharmed. Please address this critical issue.
  2. Figure 1 describes that Transcriptomic analysis reveals upregulation of RNA metabolic processes in iPSC lines after CVB3- EGFP infection. Additional figures that show the name of the genes and qPCR validation of critical genes that were differentially expressed need to be added.
  3. Figure 2 panels are missing a no-virus control. For both GFP and western blot, a negative panel and a negative lane for the cells that were not infected are required.

Author Response

General comments and general changes made to the original manuscript

The authors want to thank the reviewers for their time spent on our manuscript. Their helpful comments and suggestions for revision improved the manuscript. In Figure 3E(ii) the level of significance for mAb DAF was corrected from ** to *.

Point-by-point response to the reviewer’s suggestions

Reviewer 2

The manuscript, ‘Coxsackievirus B3 infection of human iPSC lines and 3 derived primary germ layer cells with regard to 4 receptor expression’ by Böhnke et. al., elegantly showed why enterovirus infections during human pregnancies could be involved in early pregnancy loss. This is of broad interest.

Thank you for this supportive comment.

Major comments:

#The placental barrier is historically known to restrict many viruses from entering the fetus.  The placental trophoblasts form the interface between the fetal and maternal environments are highly resistant to viral infections (Delorme-Axford et. al., 2013). The reference (31) that is cited by the authors shows that transplacental passage of virus does not occur readily and that most neonates of infected mothers will be unharmed. Please address this critical issue.

The introduction was revised and now contains additional insights from mouse models on the effect of enterovirus infection during pregnancy (line 75 to 88). Here, placental passage is indicated and a high rate of abortions and mortality was noted, especially after infection of pregnant mice at earlier stages during pregnancy. Although the rate of vertical transmission is rather low, an increase in abortion and stillbirth was noted.

#Figure 1 describes that Transcriptomic analysis reveals upregulation of RNA metabolic processes in iPSC lines after CVB3- EGFP infection. Additional figures that show the name of the genes and qPCR validation of critical genes that were differentially expressed need to be added.

We have added the names of relevant genes to the result section (line 167 to 172). The transcriptomic analysis outlined in this study is part of the already published transcriptomic analysis in the study by Bilz et al., 2018. There we have also validated hits by qPCR analysis.

#Figure 2 panels are missing a no-virus control. For both GFP and western blot, a negative panel and a negative lane for the cells that were not infected are required.

In our previous study by Hubner et al., 2017 we have assessed the effect of CVB3-EGFP infection on iPSCs in comparison to the mock-infected controls with regard to colony morphology, apoptosis induction and expression of pluripotency factors. In Figure 2 we want to highlight the effect of different plating strategies on the infection with CVB3-EGFP. In Figure 2 C and D uninfected cells were analysed. To clarify this point, we have added mock-infection to the description of Figure 2C and D.

Round 2

Reviewer 2 Report

The authors have addressed the concerns although it would have been useful to see the primary data on qPCRs and the list of genes, and that the placental barrier is indeed breached by the virus. I request the following minor change: Please add the negative controls in figure 2. This is required to show the specificity of the antibody. A western blot with one lane is hard to interprete.  Maybe the addition of the whole blot will resolve this issue.

Author Response

General comments and changes made to the original manuscript

The authors want to thank the reviewers for their time spent on our revised manuscript. In addition to their comments and suggestions we have revised the following:

  • The affiliations were updated, including the affiliations for the Institute of Virology, University of Leipzig, which was renamed to Institute of Medical Microbiology and Virology
  • Typing and grammar errors were corrected
  • Figure 1B: Oct4 and Nanong were capitalized
  • We have applied the same mode of statistical analysis (ANOVA) to our data, such that in Figure 3E(ii) the level of significance was changed according to the calculations in Figure 4.

Point-by-point response to the reviewer’s suggestions

Response to reviewer 2

#Please add the negative controls in figure 2. This is required to show the specificity of the antibody.

Regarding the specificity of our antibodies in our indirect immunofluorescence protocol, the iPSC lines used in our study were already subjected to analysis with anti-CAR antibody (clone H-300) in our previous publication (Hubner et al., 2017). This publication compared the surface expression of CAR between the two iPSC lines TMOi001-A and WISCi004-A. In our current study we have used the well-characterized clone RmcB besides the clone H-300. In the manuscript we have clarified the use of these antibodies (line 238, 273, 276, 507). In the Material & Methods section we have added “Immunofluorescence analysis of CAR with clone H-300 was done in reference to our previous publication (Hubner et al., 2017), which showed differential surface expression of CAR on TMOi001-A and WISCi004-A iPSC. This was complemented by the application of the well-characterized RmcB clone, which was also used for antibody blocking experiments. Supplement Figure 2 shows a representative flowcytometric analysis on the lack of fluorescence signal after application of the donkey anti-mouse Cy3 secondary antibody” (511 to 516).”

In the supplement section (Supplement Figure 1) we have added a representative flow cytometric analysis of the surface expression of DAF in both iPSC lines. Similar to the flow cytometric analysis of CAR expression shown in our previous publication (Hubner et al., 2017), the fluorescence signal after application of the secondary antibody alone was clearly distinct from the fluorescence signal after application of anti-DAF primary antibody together with secondary Cy3-conjugated secondary antibody. We feel flow cytometric analysis as a quantitative means to assess unspecific binding of secondary antibodies. Additionally, two different clones of anti-CAR antibodies were used in the immunofluorescence analysis in our study to assess CAR distribution within iPSC colonies.

#A western blot with one lane is hard to interprete.  Maybe the addition of the whole blot will resolve this issue.

Reviewer 2 requested control files for Figure 2. Regarding the Western blot image we have cut the PVDF membrane horizontally in pieces according to molecular weight followed by incubation with the respective antibody. There were more samples on the Western blot membrane than the two samples shown in Figure 2. The original image file is provided in Supplement Figure 1.